

# Is the reputation of *Eucalyptus* plantations for using more water than *Pinus* plantations justified?

Don A White[1,2,3] *, Shiqi Ren[1], Daniel S Mendham[4], Francisco Balocchi-Contreras[5,6], Richard P Silberstein[3,7,8] Dean Meason[9], Andrés Iroumé[10] and Pablo Ramirez de Arellano[5]

[1]Guangxi Forestry Research Institute, 23 Yongwu Road, Nanning, China

[2]Whitegum Forest and Natural Resources, PO Box 3269, Midland, WA 6056, Australia

[3]Centre for Ecosystem Management, School of Science, Edith Cowan University, Joondalup, WA 6027, Australia

[4]CSIRO Land and Water, 15 College Road, Sandy Bay, Tas. 7005

[5]Bioforest SA, Camino a Coronel km 15, Coronel, Chile, 413000

[6]Water resources and energy for agriculture PhD program, Water Resources department, Universidad de Concepción, Chillán 3812120, Chile.

[7]Hydrological and Environmental Scientific Solutions, PO Box 237, West Perth, WA 6872, Australia

[8]Agriculture and Environment, The University of Western Australia, Crawley, WA 6009, Australia

[9]Scion, Tītokorangi Drive, Private Bag 3020, Rotorua 3046, New Zealand

[10]Universidad Austral de Chile, Facultad de Ciencias Forestales y Recursos Naturales, Institute of Conservation, Biodiversity and Territory, Valdivia, Chile

*Corresponding Author – Whitegum Forest and Natural Resources, PO Box 3269, Midland WA 6056; whitegumfnrm@gmail.com

**Keywords**

vegetation evaporation efficiency, climate wetness, water balance, water use, evapotranspiration, transpiration,



**Abstract**
The effect of *Eucalyptus* plantations on water balance is thought to be more severe than for commercial
alternatives such as *Pinus* species. Although this perception is firmly entrenched, even in the scientific
community, only four direct comparisons of the effect on the water balance of a *Eucalyptus* species and a
commercial alternative have been published. One of these, from South Africa, showed that *Eucalyptus grandis*
caused a larger and more rapid reduction in streamflow than *Pinus patula*. The other three, one in South Australia
and two in Chile, did not find any significant difference between the annual evapotranspiration of *E. globulus* and
*P. radiata* after canopy closure.

While direct comparisons are few, there are at least 57 published estimates of annual evapotranspiration of either
a *Eucalyptus* or *Pinus* species. This paper presents a meta-analysis of these published data. Zhang et al. (2004)
fitted a relationship between the vegetation evaporation efficiency and the climate wetness index to published
data from catchment studies and proposed this approach for comparing land uses. We fitted the same model to
the published data for *Eucalyptus* and *Pinus* and found that the single parameter of this model did not differ
significantly between the two genera (p=0.48). This implies that for a given climate wetness index the two genera
have similar annual water use. The residuals compared to this model were significantly correlated with soil depth
for *Eucalyptus,* but this was not the case for *Pinus*. For *Eucalyptus* the model overestimates the vegetation
evaporation efficiency on deep soils and underestimates the vegetation evaporation efficiency on shallow soils.





## 1. Introduction

There are now more than 23 Mha of *Eucalyptus* plantations in the temperate and tropical zones of the world (Keenan et al., 2015; Macdicken et al., 2016). These plantations extend from near the equator to approximately 43 degrees of latitude North and South and play an important and growing role in minimizing the gap between global demand for wood products and the supply (Kanninen, 2010). These *Eucalyptus* plantations are mostly established in seasonally dry climate zones (dry tropics, sub-tropics, and Mediterranean climate types). This and the reputation of *Eucalyptus* for high rates of water use when compared to alternatives, mean that wherever large-scale planting of *Eucalyptus* has occurred, it has been associated with concern, debate and often protest about the effect of these plantations on the security of water supply (Albaugh et al., 2013). Afforestation with *Pinus* and other genera has also resulted in concern about changes in local hydrology (Huber and Iroumé, 2001; Little et al., 2009) but has not been associated with the same level of polemic or controversy as the planting of *Eucalyptus*.

In 2010, plantations managed for wood production occupied a total land area 109 Mha (Kanninen, 2010). Approximately 35% of these plantations were of *Pinus* species while 10% were *Eucalyptus* (Kanninen, 2010). The annual increase in production plantations between 2010 and 2015 was 1.2%. During this time the total area of *Pinus* plantations remained virtually unchanged and much of the global increase was in either *Eucalyptus* plantations or other short rotation options such as *Acacia* (Payn et al., 2015). The global trends in plantations are towards *Eucalyptus* or species managed on short rotations to grow pulp or biomass for energy. While these global trends are important, the conflict associated with the establishment of *Eucalyptus* plantations and the potential for reduced water availability manifests locally. In South Africa and South Australia these concerns have resulted in legislation to regulate either water use (Greenwood, 2013) or planting (Albaugh et al., 2013). The effects of *Eucalyptus* on water are currently being actively debated in Chile, where Arauco SA (the largest plantation grower in Chile and the second largest pulp producer in the world) plan to replace approximately 250,000 ha of *P. radiata* plantations with *Eucalyptus*. In China regional governments are supporting research to investigate the water benefits of mixed plantings of local species with *Eucalyptus*. It is also likely that the global goal of reduced $CO_2$ emissions will intensify debate about *Eucalyptus* water use. Given the dominance of the global plantation estates by species of *Pinus* and *Eucalyptus* and the direct substitution of *Pinus* with *Eucalyptus*, a quantitative comparison between the water use characteristics of these two genera is timely.

The evidence that plantations use more water than grasslands or dryland crops is very strong (Zhang et al., 2001; Zhang, 2004). Similarly, there is evidence that plantations use more water, and therefore generate less streamflow, than natural forest in Chile (Huber et al., 2008), Brazil (Almeida et al., 2007; Meinzer et al., 1999) and Spain (Rodriguez Suarez et al., 2014). The magnitude of the difference between plantations and natural forest is less than that observed between plantations and annual pastures (Zhang et al., 2004).

While there is a perception that *Eucalyptus* use more water than alternative commercial plantation options such as *Pinus*, three of four published comparisons of the water use (defined as evapotranspiration) reported no difference between the water use of these two genera. The evidence for higher rates of water use by *Eucalyptus* is mostly from South Africa where, in a paired catchment study, Scott and Lesch (1997) showed that, at least in the early stages of growth, *Eucalyptus grandis* W. Hill. used up to 92 mm more water per year than *Pinus patula*



Schiede ex Schltdl. et Cham. In another direct comparison of the water use of a *Pinus* and *Eucalyptus* species in
plantations, Benyon et al. (2006) found that the annual water use of plantations of *E. globulus* Labill. and *P.*
*radiata* D. Don., with or without access to shallow fresh groundwater, were not significantly different. Recent
stand and catchment scale comparisons of *P. radiata* and *E. globulus* in central Chile have found that not observed
significant differences between the average annual water use of *P. radiata* and *E. globulus* (Iroumé et al., 2021;
White et al., 2021). Given these equivocal results, and the trend towards more planting of *Eucalyptus*, it is
important to understand when and why differences might occur in the water balance of *Pinus* and *Eucalyptus*
plantations.

While it seems that the maximum rates of water use by *Eucalyptus* and *Pinus* can approach the energy limit, there
do seem to be differences between commercial *Pinus* and *Eucalyptus* in their response to soil drying. Studies in
Brazil (Lima et al., 1990) and Tasmania, Australia (Honeysett et al., 1996) have shown that when planted in deep
soils and with regular inputs of rainfall or irrigation, *Eucalyptus* plantations can use water at a rate that approaches
the energy limit. Similarly high rates of water use have also been observed in *P. radiata* plantations in southern
Australia (Benyon et al., 2006) and in Chile (Huber and Iroumé, 2001) and there are reports of rates of water use
close to the energy limit in both oil palm (Röll et al., 2015) and rubber plantations (Tan et al., 2011). Studies in
China have found that the annual rate of water use by *Eucalyptus* can be substantially less than both rainfall and
available energy (Lane et al., 2004; Ren et al., 2019). This occurs during the dry season and has also been observed
in *Pinus* species (Myers et al., 1998). Notwithstanding these similarities it has been observed that the water use
of *Pinus* species decreases more rapidly with the onset of water stress than is the case with commercial *Eucalyptus*
alternatives for the same site (Teskey and Sheriff, 1996).

Reviews of the water use potential of *Eucalyptus* have highlighted the variability of reported rates of both
transpiration and evapotranspiration (Albaugh et al., 2013; Shi et al., 2012), yet there has been no systematic
attempt to determine if the high rate of water use observed in some studies is a characteristic of *Eucalyptus* in
plantations or has more to do with the conditions that prevailed in those studies. Most of the published studies of
water balance, with a couple of exceptions (Mendham et al., 2011; Scott and Lesch, 1997) have reported water
balance measurements made within a single rotation and most studies cover only a small proportion of that
rotation. It is likely that plantations must eventually reach a long-term equilibrium with the local climate and that,
except in circumstances where trees have access to water from off-site such as a regional aquifer (see O'grady et
al. (2011b) for a meta-analysis), these high rates of water use, often observed early in the first rotation, will not
be sustained. What is needed is to determine if the longer-term equilibrium water balance of catchments planted
to *Eucalyptus* will be associated with different levels of water storage, and therefore stream flow, from that under
alternative species options for wood production plantations (Mcdonnell, 2017).

While there are only three direct comparisons of the annual water balance of *Pinus* and *Eucalyptus,* there are many
studies that quantify annual water use by either a *Eucalyptus* or a *Pinus* species. These studies, and their estimates
of water use are very situation specific. Comparison of alternative land uses is complicated by the dominant role
that climate and hydrogeology play in determining the local water balance. While vegetation cover has a smaller
effect on catchment water balance than either climate or hydrogeology it is the part of the system that can be





actively managed. If studies are available for the two genera from a comparable range of annual rainfall and
evaporative environments, then comparison might be possible through normalizing water use (evapotranspiration)
with respect to potential or energy limited evaporation and plotting this as a function of the climate wetness index
(ratio of rainfall to potential evaporation). This approach has previously been used to compare the water use of
forests with dryland agriculture (Zhang et al., 2004).

In this study, we collated published annual water balance estimates for plantations with either *Eucalyptus* and/or
a *Pinus* species, and fitted the model described by Zhang et al. (2004) to test the null hypothesis that the
evaporation characteristic of commercial *Pinus* and *Eucalyptus* plantations was not significantly different. We
also test the hypothesis that variation from this model is determined by variation in soil depth.
**2. Methods**
This paper presents a meta-analysis of published measurements of the water balance of *Eucalyptus* and *Pinus*
plantations in tropical and temperate regions. The focus of this analysis is on post-canopy closure plantations in a
notional equilibrium with the site. The behaviour of plantations is quantified by comparing an index of the function
of the crop (the vegetation evaporation efficiency, *VEE*) with an index of climate wetness in the manner proposed
by Budyko (1974) and applied by Zhang et al. (2004) to compare forests with dryland agricultural systems.
**2.1 Definitions of terms**
The terms evapotranspiration, water-use, potential evaporation, vegetation evaporation efficiency and climate
wetness index have various meanings in the scientific literature and to avoid ambiguity, they are defined here as
they are used in this paper.
**2.1.1. Evapotranspiration and water-use**
Evapotranspiration (*ET*) and water-use are used in this paper to describe total evaporation from a vegetated land-
surface. They are the sum of transpiration of all plants (*T*, the evaporation through leaf and other plant surfaces
of water drawn from the soil and transported to sites of evaporation through the xylem), water intercepted by plant
canopies and evaporated without reaching the ground (interception, *I*) and evaporation of water directly from soil
and litter (often called soil evaporation, $E_s$). All these processes are affected by the choice of crop and by the
management of that crop and should therefore be included as part of the water-use of that vegetation.
**2.1.2. Potential Evaporation (PET)**
Evapotranspiration (*ET*) by any land-use is situation specific; it is affected by the climate (energy and rainfall),
the structure and function of the vegetation and by characteristics of the soil and the litter. In this paper, for the
purposes of comparison, estimates of water-use or evapotranspiration are normalized relative to measures of the
local water supply (rainfall) and potential evaporation, which represents the energy limited maximum rate of
evaporation. There are numerous measures of reference or potential evaporation including Penman Potential
Evaporation (Penman, 1949), FAO-56 Reference Evaporation (Allen et al., 2005), Pan Evaporation and Priestley
Taylor Potential Evaporation (Priestley and Taylor, 1972). They are all intended to represent the maximum





possible rate of evaporation by a land surface covered with vegetation. In this paper, potential evaporation (*PET*)
always refers to Priestley-Taylor potential evaporation (see the notes under data analysis below to see how
Priestley-Taylor *PET* was calculated for each site). We have used the coefficient 1.26 in the Priestley-Taylor
equation; this coefficient accounts for the extra roughness of forests when compared with short crops and pastures
(Eichinger et al., 1996). The evapotranspiration of plantations may still, of course, exceed this measure of *PET*.
This may be the case if there is an additional source of energy such as advection or movement of hot air into the
forest. This might occur at the edge of a plantation, especially of it is adjacent to an area of land from which there
is a large sensible heat flux. The choice of method for calculating *PET* is less important than applying the same
method for all calculations in this analysis.

**2.1.3. Water- and energy limit, vegetation evaporation efficiency (k) and climate wetness index (CWI)**

The climate imposes limits on evapotranspiration. Evapotranspiration cannot exceed the amount of water
available which is usually limited to rainfall but may include irrigation and soil stored water and ground water
(O'grady et al., 2011a). Similarly, although evapotranspiration may exceed the calculated *PET* under some
circumstances, it is ultimately limited by available energy.
The relationship between the ratio of actual evapotranspiration to reference evaporation) and the climate wetness
index (*CWI*, the ratio of rainfall to potential evaporation) (Budyko, 1974) provides a simple way of partitioning
rainfall between evaporation and runoff. The ratio of evapotranspiration to potential evaporation is often termed
the 'evaporation efficiency' of a surface (Komatsu, 2003) and a convention has developed where the surface is
included in the name. For example, the ratio of evaporation from a soil to the potential soil evaporation is referred
to as the soil evaporation efficiency (Merlin et al., 2016). In this paper, the ratio of evapotranspiration to reference
evaporation for commercial plantations of *Eucalyptus* and *Pinus* species is referred to as their vegetation
evaporation efficiency (VEE). A more 'evaporation efficient' plantation converts a relatively greater proportion
of available energy to latent rather than sensible heat.
Zhang et al. (2004) developed a simple model that predicted vegetation evaporation efficiency (*VEE*) as a function
of the climate wetness index (*CWI*). This model is given by Equation 1 (equation A22 in Zhang et al. (2004)
below and includes the parameter *c* (an empirical catchment characteristic) which captures the effect of
hydrogeology and vegetation cover on the vegetation evaporation efficiency.
$$VEE = 1 + CWI - (1 + CWI^c)^{\frac{1}{c}}$$
Equation 1

**2.2. Meta-Analysis of Published Studies**

While direct comparisons of the water balance of *Eucalyptus* and *Pinus* plantations are few there are a reasonable
number of previously published estimates of either streamflow or evapotranspiration. These data were collated
and used in the meta-analysis described below. The studies included are described in some detail in the
supplementary material and the main features are summarised in Tables 1 and 2.  A list of potentially suitable
references were first found by conducting a series of searches of the Web of Science and Google Scholar. The
following searches were conducted:





1.    Title contains (evapotranspiration or water use) and (eucalypt or eucalyptus)
2.    Title contains (evapotranspiration or water use) and (pine or pinus)
3.    Paper contains (evapotranspiration or water use) and (eucalypt or eucalyptus)
4.    Paper contains (evapotranspiration or water use) and (pine or pinus)

The first two searches yielded less than 100 papers in total. The latter two found many thousands of articles. The
200 most relevant in each search were checked to decide their suitability. For inclusion the paper must measure
or estimate evapotranspiration by a *Eucalyptus* or *Pinus* species for at least one year. Only planted forests managed
primarily for wood production were included. Agroforestry systems were excluded as were measurements made
prior to canopy closure. Native forests and burned forests and plantations with access to the water table were also
excluded. Several of the studies covered multiple years. A single value of rainfall and evaporation was calculated
as the average of all the years in each study. Sometimes a paper reported multiple estimates of evapotranspiration
for forests in the same location and growing under the same conditions. In these cases, average values were
calculated for the multiple sites.

After applying these criteria to articles found in the above searches, a total of 30 *Pinus* and 27 *Eucalyptus* stands
were included in the meta-analysis. The location, rainfall data and evapotranspiration data are provided as
supplementary material. The estimates of evapotranspiration were made using one of four methods. The method
applied in each study is indicated in Table 1.

### 2.2.1. Method 1 – Measurement and addition of component fluxes

At the stand or plot scale evapotranspiration (water-use) is the sum of evaporation from the soil and leaf litter ($E_s$),
evaporation of rainfall intercepted by the vegetation canopy ($I$) and transpiration or the direct uptake of water by
the trees and the evaporation of this water through the leaf surface ($T$). Evapotranspiration can therefore be
calculated as the sum of the component processes.

### 2.2.2. Method 2 – One dimensional water balance

Provided there is no leakage or runoff then evapotranspiration (*ET)* can be calculated in stand scale studies as the
sum of rainfall ($P$) and the change in the soil water content ($\Delta S$) between two measurements.

$ET = P + \Delta S$   Equation 2.

### 2.2.3 Method 3 – Catchment water balance

For a catchment, if there is no change in the amount of water stored in the soil or the groundwater ($\Delta S$),
evapotranspiration (*ET*) is simply the difference between rainfall and streamflow ($Q$). Over long time periods it
is often assumed that the change in storage is negligible; this is less valid as the period of the estimate is reduced
or if the annual total rainfall has a clear temporal trend.

$ET = Q - P + \Delta S$ Equation 3



### 2.2.4 Method 4 – Eddy covariance (flux towers)

Properly located flux towers can be used to estimate the net carbon and water flux (evapotranspiration) above an ecosystem. The instruments on these towers measure the total solar and net radiation and partition this to latent (evapotranspiration) and sensible heat flux (air temperature change) and heat storage changes in soil and biomass. The covariances of high frequency measurements of air temperature, humidity and $CO_2$ are used to calculate total evaporation and carbon exchange between the atmosphere and the underlying vegetation (Aubinet et al., 2012). Measurements are typically made on a 30-minute time interval to represent fluxes from an upwind surface area or "footprint". The area of the footprint is dependent on strength of the turbulence in the air, a function of wind speed and surface roughness elements, and the height of the instruments, thus the location of land surface influencing the measurements changes through time. Eddy covariance measurements give total fluxes from the contributing footprint, thus are useful for total ecosystem energy, water and carbon balances. However, partitioning the fluxes between different contributing vegetation and soil components requires additional measurements, such as sap flow, rain throughfall and soil evaporation. Also, the measurements are unreliable during periods of stable air and low turbulence, such as still cold nights but, for the purposes of the analyses in this paper, these are periods typically with very low water fluxes and have only minor influence on the total system water balance. There is a substantial literature describing these methods and complementary measurements, a detailed description is beyond the scope of this paper but can be found in Wilson et al. (2001) where the method is compared with alternatives.

### 2.3. Variations at Two Sites

A study by Scott and Lesch (1997) at Mokobulaan in South Africa reported more rapid changes in streamflow after planting of *E. grandis* than after planting of *P. patula*. The soil was very deep, and it is probable, though this was not measured, that evapotranspiration exceeded rainfall and that this was more pronounced in the *E. grandis* than the *P. patula*. To allow for this effect we assumed a storage of 100 mm per metre of soil and a rate of root extension of 2 m per year for *E. grandis* after (Dye, 1996) and 1 m per year in *P. patula*. This relative rate is consistent with the observation that streamflow ceased 5 and 10 years respectively, after planting of *E. grandis* and *P. patula* (Scott and Lesch, 1997).

Another study included here was made at Lewisham in Tasmania, Australia by Honeysett et al. (1996). In this study the effect of drought on the water relations and water balance of *E. globulus* and *E. nitens* were investigated using irrigated controls and rainfed plots. The irrigated treatments were excluded from this meta-analysis. However, to avoid mortality the rainfed treatments received some supplementary irrigation. This irrigation is included in the rainfall figure in Table 1 and in the supplementary material.

### 2.4. Derived climate and vegetation indices

In each of the papers included in this analysis, evapotranspiration (*ET*) was estimated from the measurement of other variables by one of the four methods described above. Rainfall data was available for all the studies included in this review. Time series climate data from the 0.5-degree grid point closest to each site was also downloaded for the duration of each experiment (Climate Research Unit Time Series v4.03, Harris et al., 2014). Net radiation





was calculated for the location after Hargreaves and Samani (1985.) and then Priestley-Taylor evaporation (*PET*)
was calculated as:
$$\lambda PET = 1.26 \left[ \frac{s}{s+\gamma} \right] R_n \qquad \text{Equation 4}$$
where $R_n$ is net radiation in W m$^{-2}$, $\lambda$ is the latent heat of vapourisation of water (2245 kJ kg$^{-1}$), $s$ is the slope of
the relationship between saturated vapour pressure and temperature (kPa °C$^{-1}$) and $\gamma$ is the psychrometric constant
(kPa °C$^{-1}$). These 'constants' are temperature dependent; $s$ was calculated using the empirical model in Equation
5 (Hahn and Landeck, 1998.) and $\gamma$ was calculated using Equation 6 in which $T_a$ and $P_a$ are average daily air
temperature (calculated as the average of $T_{max}$ and $T_{min}$) and atmospheric pressure (assumed to be 101.3 kPa), $c_p$
is the specific heat of dry air (1.013 kJ kg °C$^{-1}$ ) and $\varepsilon$ is the ratio of the molecular weight of water to dry air
286  (0.622).

$$s = 0.04145 e^{0.06088 T_a} \qquad \text{Equation 5}$$
$$\gamma = \frac{c_p P_a}{\lambda \varepsilon} \qquad \text{Equation 6}$$
For each measurement year at each study location the vegetation evaporation efficiency (*VEE*) and the climate
wetness index were also calculated using equations 7 and 8 respectively.
$$VEE = \frac{ET}{PET} \qquad \text{Equation 7}$$
$$CWI = \frac{P}{PET} \qquad \text{Equation 8}$$
**2.5. Meta-Analysis**
The values of the vegetation evaporation efficiency estimated from each of the published studies were plotted as
a function of the climate wetness index. The model described in Equation 1 was then fitted to the data using the
Nonlin function in *R* and the parameter *c* and the coefficients of determination, $r^2$, value were calculated for each
genus separately and for the pooled data (R-Core-Team, 2013). Analysis of variance was also completed to test
for a significant difference between *Pinus* and *Eucalyptus* in the parameter *c* (R-Core-Team, 2013). The residuals
(predicted minus observed) were plotted against soil depth for the sites where this data was available. Linear
regression was used to explore the relationship between annual transpiration and annual evapotranspiration.
Simple t-tests for non-paired observations were used to test for differences between genera in annual
evapotranspiration and the ratio of evapotranspiration to rainfall.



**3. Results**
**3.1. Rainfall Limited Plantations**
Twenty-seven *Eucalyptus* and 30 *Pinus* sites were included in the meta-analysis. The details of these sites are
summarized in three tables. The most detailed information is in the supplementary material together with the
measured and calculated climatic data, estimated evapotranspiration, and the detailed results of the data analysis.
The papers from which the data were taken are listed in Table 1 with the rainfall data, species studied, and the
method used to estimate evapotranspiration. Table 2 summarises the range of climatic conditions and evaporation
rates by species and indicates the number of studies for each species by country or continent.

The analysis included sites from tropical, dry tropical, sub-tropical, warm temperate, cool temperate,
Mediterranean, and montane climates with both genera represented in all but one climate type and in most
locations. There is a bias of *Pinus* studies to the United States and of *Eucalyptus* to Australia (Table 2). Species
of *Eucalyptus* represented in order of decreasing number of estimates were *E. globulus* (10), *E. nitens* (H. Deane
& Maiden) Maiden (7), *E. urophylla* S.T. Blake (3), *E. grandis* (2), *E. urophylla* x *grandis* (2), *E. urophylla* x
*globulus* (2) and *E. saligna* Sm. (1) (Table 1). Similarly estimates for species of *Pinus* were made for *P. radiata*
(18), *P. taeda* L. (5), *P. patula* (2), a mixed stand of *P. taeda* and *P. palustris* Miller (1), mixed stand of *P. elliottii*
Engel. and *P. palustris* (1), *P. elliottii* (1), *P. caribaea* var hondurensis W.H. Barrett and Golfari (1) and *P. strobus*
L. (1) (Table 1). Thus, each genus is represented by species from tropical, sub-tropical and temperate
environments.
**3.1.1. Annual Rainfall and Evapotranspiration**
The annual rainfall at the 24 *Eucalyptus* sites ranged from 489 mm at one of the South Australian sites to 2088 mm
at a site in the Rio Grande du Sol in Southern Brazil. The range of rainfall was similar for the 27 *Pinus* sites and
varied from 600 mm, at a South Australian site to 2081 mm at a site near Valdivia in south central Chile.
Interestingly, both the low rainfall site in South Australia and the high rainfall site in Chile were planted to *P.*
*radiata*. The situation was similar for average annual potential evaporation which ranged from 1005 to 2008 mm
at the *Eucalyptus* sites and from 1021 to 2004 mm at the *Pinus* sites (supplementary material). The median annual
rainfall for the *Eucalyptus* and *Pinus* sites respectively was 940 mm and 927 mm while average potential
evaporation was 1480 mm and 1551 mm (Table 2). Thus, the range and median conditions covered by the sites
included in this meta-analysis was very similar for both genera.

Annual evapotranspiration increased as a function of rainfall before plateauing in the same manner as reported by
Zhang et al. (2001). Annual rates of evapotranspiration reported for *Eucalyptus* species were between 488 mm at
a low rainfall site in South Australia planted to *E. globulus* (Benyon et al., 2006) and 1345 mm at a site in Brazil
planted to *E urophylla x E. grandis* (Soares and Almeida, 2001). The lowest and highest annual evapotranspiration
for *Pinus* species were 355 mm for *P. radiata* at Jonkershoek in the Western Cape of South Africa (Lesch and
Scott, 1997) and 1291 mm for *P. strobus* in North Carolina (Ford et al., 2007).

The minimum, mean, median and maximum rates of evapotranspiration were all slightly greater for the *Eucalyptus*
sites than for the *Pinus* sites (Figure 1). This, albeit non-significant (p=0.24), difference was associated with the





*Eucalyptus* sites generally being on slightly wetter sites. When evapotranspiration was divided by rainfall the
median values of the ratio for the two genera were nearly identical at 0.77 and 0.76 (Figure 2). The ratio of
evapotranspiration to rainfall varied from 0.45 to 1.31 in *Eucalyptus* and from 0.44 to 1.2 in *Pinus* species. At one
site in South Africa (Lesch and Scott, 1997) the rate of evapotranspiration by *E. grandis* exceeded rainfall by 31%
(Figure 2).  At the same site, evapotranspiration by *P. patula* exceeded rainfall by 19% (Figure 2).

### 353 3.1.2. Vegetation evaporation efficiency as a function of the climate wetness index (*Eucalyptus* and *Pinus*)

In Figure 3 the vegetation evaporation efficiency for each study site is plotted as a function of the climate wetness
index. For both the *Eucalyptus* and *Pinus* sites there is a strong, positive correlation between the vegetation
evaporation efficiency and the climate wetness index. For the *Eucalyptus* sites the model of Zhang et al. (2004)
(Equation 1) explained 66 % of the variation in the vegetation evaporation efficiency while for *Pinus* this
decreased to 63 %. The parameter $c$ in the model described by Equation 1 fitted to the data was 2.84 for *Eucalyptus*
and 2.64 for *Pinus*. While this may be an important difference it was not statistically significant (p=0.50) and the
value for $c$ when the relationship was fitted to the pooled data was 2.74 and the $r^2$ was 0.69. Figure 4 shows the
ratio of the predicted vegetation evaporation efficiency for *Eucalyptus* to the predicted vegetation evaporation
efficiency for *Pinus* as a function of climate wetness index. The maximum proportional effect of genus on the
vegetation evaporation efficiency of 3.5% is predicted to occur where the Climate Wetness Index is 1.

### 364 3.1.3. The effect of soil depth

While the relationships in Figure 3 are significant for both genera there is nonetheless substantial scatter. The soil
depth was not provided in all the papers included in this analysis. When the residuals (observed minus predicted)
were plotted as a function of the soil depth the relationship was significant for the *Eucalyptus* sites (Figure 5) but
not for the *Pinus* sites (data not shown). A linear relationship with soil depth explained 57% of the error for
*Eucalyptus* and indicated that the model shown in Figure 3, for c of 3.1, overestimated the vegetation evaporation
efficiency in shallow soils and underestimated it in deep soils (Figure 5), with the model having zero residual with
a soil depth around 10 m.

### 372 3.1.4. Transpiration as a proportion of evapotranspiration

A subset of the studies, again indicated in the supplementary material, also provided estimates of transpiration
made using sapflow sensors. For both *Eucalyptus* and *Pinus* there was a strong linear relationship between
transpiration and evapotranspiration with an approximate slope of 0.5 (Figure 6).

### 376 4. Discussion

The results of the meta-analysis of published records of evapotranspiration for *Eucalyptus* and *Pinus* species in
this paper suggest that for a given climate wetness index the water use of *Eucalyptus* and *Pinus* plantations is not
significantly different (p=0.50). This does not mean that there are not circumstances, or periods within a rotation,
when *Eucalyptus* will use more water than the alternatives. The water balance of plantations and alternatives is
very situation specific, and our focus should be on understanding the sources of variation rather than generalizing
about one land use option. The work of Scott and Lesch (1997) and the results of White et al. (2009) from three





*E. globulus* plantations established in south-western Australia highlight the potential of *Eucalyptus* plantations to
exceed the water limit early in the rotation on deep soils. This is an issue that warrants deeper understanding and
the development of management strategies. The results of the meta-analysis suggest that the average annual water
use by the two genera will be similar over large areas and long time periods (decades). They do not, however,
preclude periods of high-water use by *Eucalyptus*.

The range of annual rainfall, climate wetness indices and annual evapotranspiration in the published studies was
similar for the 27 *Eucalyptus* and 30 *Pinus* sites included in meta-analysis (Table 1, Table 2 and supplementary
material).  Only a few sites had climate wetness indices more than 1.5. These were Jijou and Hetou in China,
Huape and Valdivia in central Chile and Coweeta in North Carolina. In the case of the Chinese sites, Lane et al.
(2004) and Ren et al. (2019) concluded that plantations of *Eucalyptus* would not have an important effect on water
resources nor on water security in this part of China. Notwithstanding this conclusion there is still a lot of
investment made to quantify to water use of *Eucalyptus* in these regions. Wherever the climate wetness index
exceeds 1.5 then the amount of streamflow will always be substantial, even in lower rainfall years (White et al.,
2016). Thus, rather than annual water balance, the focus should be on water quality and dry season flow to better
understand the effect of land use change, including the planting of *Eucalyptus*, on water security.

For the published *Eucalyptus* and *Pinus* studies analysed here, there was a strong positive correlation between
evapotranspiration and rainfall and therefore between the vegetation evaporation efficiency and the climate
wetness index (Figure 3). The coefficient, or ´catchment characteristic´, $c$ was greater in *Eucalyptus* (2.84) than
in *Pinus* (2.64) but the difference between the two genera was not statistically significant (p=050). When this
result was discussed with colleagues in the forestry sector or with people in the forest research community it met
with responses ranging from mild surprise to disbelief. The belief that *Eucalyptus* uses more water than any of
the alternative crops is very firmly entrenched even though it does not seem to have a firm scientific foundation.
Given that the meta-analysis presented in this paper produced a result that was counter to the prevailing view it is
very important to consider the direct and corroborative evidence that either support or oppose this observation.
The following paragraphs attempt to provide a mechanistic basis for the observation that, while under some
circumstances *Eucalyptus* can use water much more rapidly than *Pinus* (Scott and Lesch, 1997), the average
behaviour of the two genera appears similar (Benyon and Doody, 2015), Figure 3. This mechanistic basis is then
used to indicate under which circumstances the effects of plantations of *Pinus* or *Eucalyptus* species on water
resources should be evaluated and actively managed.

The key to understanding the patterns of water use in *Eucalyptus* and *Pinus* plantations lies in the hydraulic
architecture of the two genera and in the way that this affects the relationship between water use and carbon gain.
There are some consistent differences between the group of *Eucalyptus* and *Pinus* species that are grown in
commercial plantations. First, and very importantly, *Pinus* species are gymnosperms and their water conducting
elements are tracheids while in *Eucalyptus* water is transported in vessels. The maximum hydraulic conductivity
of angiosperms exceeds that of conifers with almost no overlap in the ranges (Sperry et al., 2006). It is the diameter
of the vessels that afford angiosperms greater maximum hydraulic conductance (Sperry et al., 2006). It is also
known that in the *Eucalyptus* genus vessel size, and maximum hydraulic conductivity of the xylem, is correlated





with climate wetness (Pfautsch et al., 2016) so that the major plantation species can have hydraulic conductivities
among the highest in the plant kingdom. Leaf conductance and maximum photosynthetic capacity scale directly
with the hydraulic conductivity of the xylem (Hubbard et al., 2001; Tyree, 2003).

Thus, plantation *Eucalyptus* species, the most important of which are from the Symphyomyrtus subgenus and
grow naturally in the wetter fringes of the Australian continent, have higher maximum hydraulic conductivity,
water use and photosynthetic capacity than commercially grown *Pinus* species (Whitehead and Beadle, 2004). In
the early growth phase, Symphyomyrtus *Eucalyptus* species also have a much higher specific leaf area (ratio of
leaf area to mass) than *Pinus* and this results in more rapid canopy development and the potential for faster early
growth and water use such as observed by Scott and Lesch, (1997). This can of course only happen if there is
water available to support this growth and canopy development and this can be supplied by rainfall throughout
the year or by additional sources of water stored in deep soil (Dye and Olbrich, 1992; Scott and Lesch, 1997;
White et al., 2014), shallow groundwater (Benyon et al., 2006; Brooksbank et al., 2011; Eamus et al., 2000;
O'grady et al., 2011b) or applied as irrigation (Honeysett et al., 1996). If *Eucalyptus* plantations are grown on
deep soils and in regions where the climate wetness index is much less than one (potential evaporation exceeds
rainfall) then, by virtue of their hydraulic architecture, they have the potential to affect the water balance more
than alternatives.

The capacity of *Eucalyptus* for high instantaneous sap velocities that are associated with elevated photosynthetic
capacity also affects the seasonal patterns of water use in *Eucalyptus* compared to *Pinus*. Transpiration of
*Eucalyptus* species increases rapidly in spring associated with high maximum stem and leaf conductivity (White
et al., 1999). The relative behaviour of *E. globulus* and *P. radiata* is well understood making them good exemplars.
They are also two plantation species of great global importance that are grown in similar areas including in central
Chile. In Chile and Australia, *P. radiata* is known to be capable of surviving more severe droughts than *E. globulus*
and plantations of the species therefore extend into drier areas than *E. globulus* both in Chile and in Australia.
The greater drought tolerance of *P. radiata* than *E. globulus* is mediated by a much stronger stomatal response to
soil drying (Mitchell et al., 2014). In situations where the amount of soil water storage imposes an upper limit on
annual use then, while this store of water will be completely depleted by both species, it will be used earlier in the
growing season by *Eucalyptus*. Thus, the period of peak physiological activity and growth in *Eucalyptus* is
associated with lower average temperatures and more moderate air saturation deficits. This pattern of water use
biased towards spring and early summer can result in very efficient water-use growth and wood production (White
et al., 2015). This behaviour of the *Eucalyptus* is closer to a mimic of the seasonal water use pattern of an annual
species. This mechanism underlies the greater water use efficiency of *Eucalyptus* species than of the *Pinus* but is
also associated with an increased risk of mortality (White et al., 2003; White et al., 2009) if the soil water runs
out. It also underlies the high rates of water use sometimes observed on deep soils (Scott and Lesch, 1997).

At equilibrium *Eucalyptus* and *Pinus* species generally have different seasonal patterns of water use. Nonetheless,
the average annual water use does not differ significantly between the two genera amongst the published studies
presented in Figure 3. This observation is entirely consistent with the observed hydraulic architecture of these two
genera in the field. Radiation interception and absorption, and therefore productivity and evapotranspiration in


forests, including plantations, are strongly correlated with leaf area index. Battaglia et al. (1998) proposed that after the canopy closes, plantations will arrive at an 'equilibrium' leaf area index that maximises the net primary productivity. They further demonstrated that the value of this optimum leaf area index is strongly influenced by the climate wetness; higher optimum values of leaf area index were observed in wetter situations. The value of this 'optimum leaf area index' tends to be higher for a given climate wetness in *Pinus* species than in *Eucalyptus* species. For those experiments included in this analysis that reported leaf area index, the average value for *Pinus* was approximately 4, nearly a full unit greater than the average value for the *Eucalyptus* plantations.

In comparing *Eucalyptus* and *Pinus* in commercial plantations it is important to compare at least one and possibly more, full crop rotations. *Pinus* is generally managed for solid wood production and therefore on a longer rotation than *Eucalyptus* which is usually, but not exclusively, grown for pulpwood production. Around the world the time from planting to harvest of *Pinus* species is between two and three times that of the *Eucalyptus* in the same location. In Chile, for example, *Eucalyptus* is harvested after about 12 years while *Pinus* is grown for about 25 years. *Pinus* is usually grown for solid wood or veneer production and is therefore thinned at least once and is often pruned to produce clear wood. After the harvesting of the first *Eucalyptus* crop, a *Pinus* plantation on the same location would remain standing and operating at, or near, the water limit. For a period of between two and three years after the *Eucalyptus* harvest the evapotranspiration of the *Pinus* will therefore exceed that of the *Eucalyptus*. This is evident in the results of Scott and Lesch (1997) who compared *E. grandis* with *P. patula*. The frequency of harvest of *Eucalyptus* will be a key factor affecting the comparative water balance of *Pinus* and *Eucalyptus* plantations. Paradoxically, more frequent harvests will increase the average streamflow from *Eucalyptus* plantations relative to *Pinus*. It has been demonstrated that the effects of thinning on the water balance are transient, lasting for a maximum of one year in both *Pinus* and *Eucalyptus* (Scott and Lesch, 1997; White et al., 2014).

The proportion of evapotranspiration that occurs as transpiration was approximately 0.5 for both *Pinus* and *Eucalyptus* across a wide range of climate wetness indices (Figure 6). This means that the annual partitioning of evapotranspiration to fluxes other than transpiration is similar for these two genera. The partitioning of these other fluxes to understorey transpiration, soil evaporation and interception may have important implications for ecosystem productivity and efficiency. The water use efficiency of wood production is directly correlated with the ratio of transpiration to other fluxes (White et al., 2015). In a study that compared *E. globulus* and *P. radiata* Benyon and Doody (2015) observed that interception was more than half the non-transpirational fluxes in *P. radiata* and less than half in *E. globulus*. This variation in partitioning is a direct consequence of the previously noted tendency for *Pinus* to have a higher leaf area index than *Eucalyptus* and the greater canopy storage per unit leaf area in *Pinus* than in broadleaved species (Iida et al., 2005). A weakness of this analysis and of the literature on water balance is the exclusion of stemflow from most water-balance studies. It is likely that stemflow will contribute more to throughfall in *Eucalyptus* (7% of rainfall) than in *Pinus* (2 to 5%) (Crockford and Richardson, 1990). This difference is approximately equivalent in magnitude to the observed, albeit non-significant, difference between the genera in this analysis.

**5. Conclusion**



Water use by vegetation is very situation specific. The comparison between *Eucalyptus* and *Pinus* depends on the age of the plantation, the length of the rotation, the seasonality of rainfall and the depth of the soil. In this paper a meta-analysis of published estimates of evapotranspiration by *Pinus* and *Eucalyptus* species in commercial plantations did not find a significant difference between the genera. Specifically, while there was a small, but systematic difference of about 3% in water use between the genera (see Figures 5 and 6), this analysis finds that for a given climate wetness index the evapotranspiration by *Pinus* and *Eucalyptus* was statistically the same. Moreover, our understanding of the hydraulic architecture and stomatal physiology of pines and eucalypts suggests that, although the long-term average behaviour may be similar, there will be differences in their temporal pattern of water use both within and between years. *Eucalyptus* will use more water than *Pinus* early in the growing season and in the early years of the rotation. On deep soils this may result in lasting differences but under most circumstances the total effect on water balance will be similar. The reputation of much higher water use by *Eucalyptus* may stem partly from the observation of vigorous early growth of *Eucalyptus* and the many studies on young plantation stands.

**Competing Interests**

From July 2015 to April 2020, Drs White and Silberstein were paid to provide advice to Bioforest SA on Ecohydrology and Ecophysiology. Bioforest SA are an R and D company owned by Arauco, the largest plantation grower in central Chile. In the course of this work Dr White has also received some financial support from the Guangxi Forestry Research Institute in China.

**Code / Data Availability**

Provided as Supplementary Material

**Author Contributions**

Don A White – Conceptualization, Data Curation. Formal Analysis, Methodology, Validation, Original Draft Preparation, Review and Editing

Shiqi Ren – Conceptualisation, Funding acquisition, Supervision

Daniel Mendham – Conceptualisation, Data Curation, Formal Analysis, Review and Editing

Francisco Balocchi-Contreras - Conceptualisation, Review and Editing

Richard Silberstein – Conceptualisation, Review and Editing

Andrés Iroumé – Conceptalisation, Validation

Pablo Ramirez de Arellano – Conceptualisation, Methodology, Project Administration, Supervision



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





Table 1. Brief description of all the papers and the associated studies included in the meta-analysis. See the supplementary material for a full summary of the data used in the analysis. Data are sorted by Region and Annual Rainfall. The annual rainfall data provided here are measurements from the cited paper unless noted otherwise.

| Species | Region | Number of Sites | Rainfall Range (mm) | Climate Type | Number of Years Data | Method Used to Estimate ET | Reference |
|---|---|---|---|---|---|---|---|
| *E. urophylla x globulus* | Terra Dura, Brazil | 2 | 1433 - 1626 | Sub-Tropical | 12* | Method 3 | (Almeida et al., 2016) |
| *E. globulus* | Green Triangle, Australia | 3 | 489-701 | Cool Temperate | 3 to 4+ | Method 1 | (Benyon et al., 2006) |
| *E. globulus* | Portugal | 2 | 788 | Mediterranean | 9# | Method 3 | (David et al., 1994) |
| *E. globulus* | Tasmania, Australia | 1 | 975 | Cool Temperate | 4# | Method 2 | (Honeysett et al., 1996) |
| *E. nitens* | Tasmania, Australia | 1 | 960 | Cool Temperate | 4# | Method 2 | (Honeysett et al., 1996) |
| *E. urophylla* | *Leizhou Peninsula, China* | 2 | 1620-1920 | Tropical | 2+ | Method 1 | (Lane et al., 2004) |
| *E. grandis* | Northern Province, South Africa | 1 | 756 | Sub-Tropical | 9+ | Method 3 | (Lesch and Scott, 1997) |
| *E. urophylla x grandis* | Grao Mogol, Brazil | 1 | 1121 | Tropical | 2+ | Method 2 | (Lima et al., 1990) |
| *E. saligna* | Rio Grande du Sol, Brazil | 1 | 2088 | Sub-Tropical | 1+ | Method 3 | (Reichert et al., 2017) |
| *E. urophylla* | Guangxi, China | 1 | 1294 | Sub-Tropical | 1# | Method 1 | (Ren et al., 2019) |
| *E. nitens* | Tasmania, Australia | 4 | 1222-1259 | Cool Temperate | 1-3# | Method 1 | (Roberts et al., 2015) |
| *E. globulus* | South India | 1 | 1568 | Montane | 9* | Method 3 | (Samraj et al., 1988) |
| *E. grandis* | South Africa | 1 | 1163 | Sub-Tropical | 10# | Method 3 | (Scott and Lesch, 1997) |
| *E. urophylla x grandis* | Aracruz, Brazil | 1 | 1396 | Tropical | 1+ | Method 2 | (Soares and Almeida, 2001) |
| *E. globulus* | Arauco, Chile | 1 | 1395 | Mediterranean | 3 | Method 1 | (White et al, 2021) |
| *E. nitens* | Curanilahue, Chile | 2 | 1845 | Mediterranean | 3 | Method 2 | (Balocchi et al., 2020) |
| *E. globulus* | Nascimiento, Chile | 2 | 1272 | Mediterranean | 8 | Method 1 | (Iroumé et al., 2021) |





| | | | | | | | |
|---|---|---|---|---|---|---|---|
| *P. taeda and P. palustris* | South Carolina, USA | 1 | 1319 | Sub-Tropical | 20+ | Method 3 | (Amatya et al., 2006) |
| *P. radiata* | New Zealand | 1 | 1554 | Cool Temperate | 27* | Method 3 | (Beets and Oliver, 2006) |
| *P. radiata* | Green Triangle, Australia | 4 | 600-724 | Cool Temperate | 4+ | Method 1 | (Benyon et al., 2006) |
| *P. radiata* | NE Victoria, Australia | 1 | 1400 | Cool Temperate | 1+ | Method 3 | (Bren and Hopmans, 2007) |
| *P. elliottii* | SE Queensland, Australia | 1 | 1284 | Sub-Tropical | 10+ | Method 3 | (Bubb and Croton, 2002) |
| *P. strobus* | North Carolina, USA | 1 | 2240 | Sub-Tropical | 2+ | Method 1 | (Ford et al., 2007) |
| *P. taeda* | Florida, USA | 2 | 1098-1175 | Tropical | 2-4# | Method 4 | (Gholz and Clark, 2002) |
| *P. radiata* | Central Chile | 4 | 1084-2081 | Mediterranean | 2-3+ | Method 1 | (Huber and Iroumé, 2001) |
| *P. radiata* | Western Cape, South Africa | 1 | 642 | Mediterranean | 11# | Method 3 | (Lesch and Scott, 1997) |
| *P. patula* | Natal, South Africa | 1 | 886 | Sub-Tropical | 11# | Method 3 | (Lesch and Scott, 1997) |
| *P. caribea var hondurensis* | Grao Mogol, Brazil | 1 | 1121 | Tropical | 3 | Method 2 | (Lima et al., 1990) |
| *P. elliottii and P. palustrus* | North Carolina, USA | 2 | 883-1033 | Sub-Tropical | 4 | Method 4 | (Powell et al., 2005) |
| *P. radiata* | Central Tablelands, NSW, Australia | 1 | 738 | Cool Temperate | 16 | Method 3 | (Putuhena and Cordery, 2000) |
| *P. patula* | Northern Province, South Africa | 1 | 756 | Sub-Tropical | 17 | Method 3 | (Scott and Lesch, 1997) |
| *P. taeda* | North Carolina, USA | 1 | 1091 | Sub-Tropical | 4 | Method 4 | (Stoy et al., 2006) |
| *P. taeda* | North Carolina, USA | 1 | 1238 | Sub-Tropical | 4 | Method 4 | (Sun et al., 2010) |
| *P. radiata* | Constitucion, Chile | 1 | 1016 | Mediterranean | 1 | Method 1 | (White et al., 2021) |
| *P. radiata* | Arauco, Chile | 1 | 1395 | Mediterranean | 3 | Method 1 | (White et al., 2021) |
| *P. radiata* | Valdivia, Chile | 2 | 2210 | Mediterranean | 8 | Method 2 | (Balocchi et al., 2020) |





| *P. radiata* | Nascimiento, Chile | 2 | 1272 | Mediterranean | 8 | Method 1 | Iroumé et al. (2021) |
|---|---|---|---|---|---|---|---|

*Full Rotation

+Post Canopy Closure Only

#Includes Pre and Post Canopy Closure

Table 2. Summary of the studies included in the meta-analysis (see Appendix for more details, and references for each study). This table indicates the number of studies included by country or continent, species, and climate zone.

| | | Eucalyptus | Pinus | Total |
|---|---|---|---|---|
| **Country/Continent** | Australia and New Zealand | 9 | 8 | 17 |
| | United States | 0 | 8 | 8 |
| | South America | 10 | 11 | 21 |
| | South Africa | 2 | 3 | 5 |
| | China | 3 | 0 | 3 |
| | Europe | 2 | 0 | 2 |
| | India | 1 | 0 | 1 |
| | Total | 27 | 30 | 57 |
| **Rainfall (mm) and Evapotranspiration (mm)** | Min Annual Rain | 489 | 600 | |
| | Median Annual Rain | 1259 | 1152 | |
| | Max Annual Rain | 2088 | 2240 | |
| | Min Annual ET | 488 | 355 | |
| | Median Annual ET | 940 | 927 | |
| | Max Annual ET | 1345 | 1291 | |





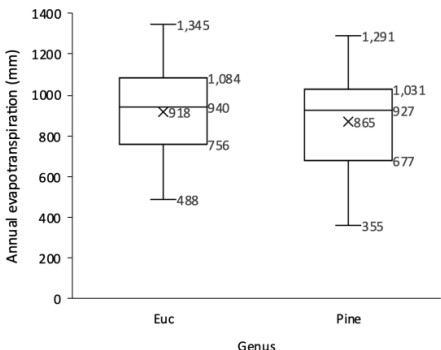

Figure 1. Box and whisker plots of annual evapotranspiration for the *Eucalyptus* and the *Pinus* sites. The three horizontal lines in the box show the median, 25th and 75th percentile values. The whiskers show the minimum and maximum values and the x indicates the mean. The associated labels indicate the actual values.

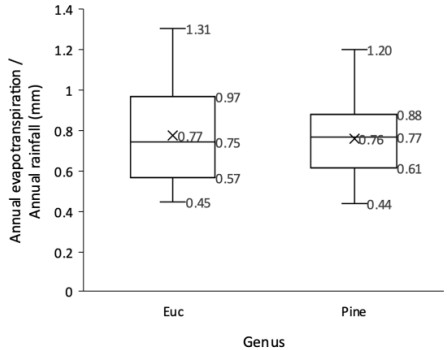

Figure 2. Box and whisker plots of the ratio of the evapotranspiration to rainfall for the *Eucalyptus* and the *Pinus* sites. The three horizontal lines in the box show the median, 25th and 75th percentile values. The whiskers show the minimum and maximum values, and the x indicates the mean values. The associated labels indicate the actual values. The mean ratio was 0.81 for *Eucalyptus* and 0.79 for *Pinus* while the medians for the same two genera were 0.77.





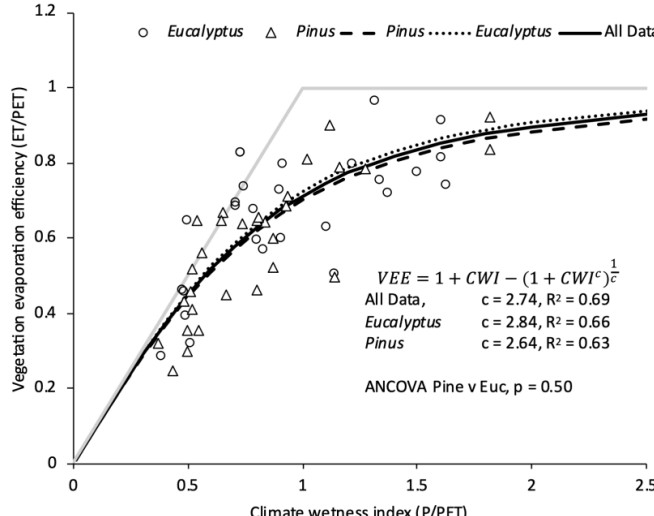

Figure 3. The vegetation evaporation efficiency as a function of the climate wetness index (a Budyko plot) for 57 (27 *Eucalyptus* and 30 *Pinus*) published studies. The solid grey lines are the water limit (evapotranspiration is equal to rainfall) and the energy limit (evapotranspiration is equal to potential evaporation). The dotted and dashed lines are for Equation 1 fitted separately to the data for *Eucalyptus* and *Pinus*.

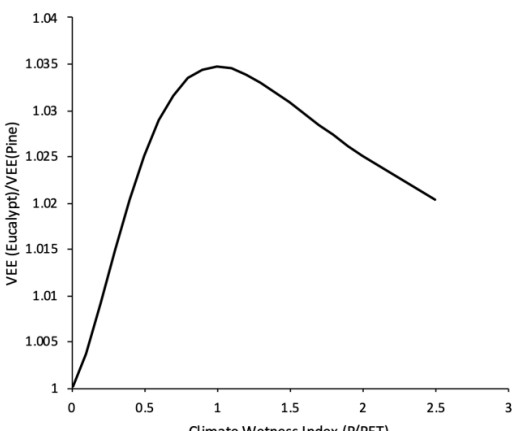

Figure 4. The ratio of the vegetation evaporation efficiency (*VEE*) for *Eucalyptus* to the vegetation evaporation efficiency for *Pinus* plotted as a function of the Climate Wetness Index. The vegetation evaporation efficiency was predicted using the separate relationships for the two genera in Figure 3.

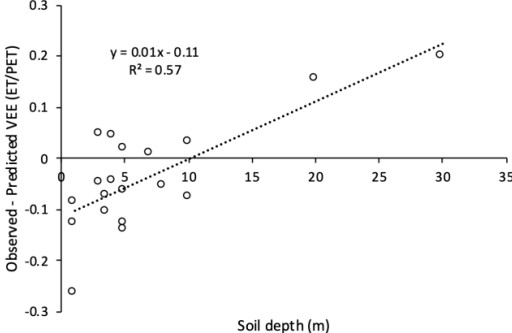

Figure 5. The residuals from Figure 4 for the *Eucalyptus* sites plotted as a function of soil depth. The model in Figure 4 with a value for c of 3.1 overestimates the observed value of k in shallow soils and underestimates k in deep soils.

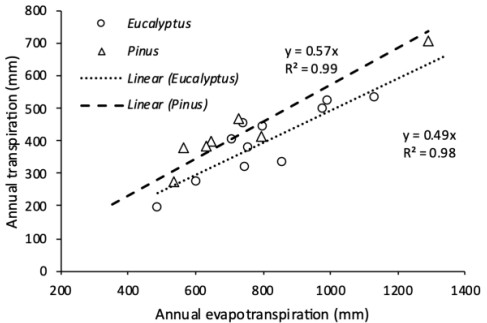

Figure 6. The relationship between annual transpiration and annual evapotranspiration for the subset of sites where transpiration was measured using sapflow sensors.