# Peer review of "Is the reputation of *Eucalyptus* plantations for using more water than *Pinus* plantations justified?"

_Hydrology and Earth System Sciences, 2022_

## Author Response (AR1)

Authors Response to Reviewer Comments on hess-2022-200

Is the reputation of *Eucalyptus* plantations for using more water than *Pinus* plantations justified?

Don White – on behalf of all authors.

I thank both reviewers for taking the time to review our paper. Their comments were constructive and provided an opportunity to improve the paper. The following is a 'comment by comment' response to each review, providing a detailed indication of any changes made to the paper

Reviewer 1

Comments:

1. A regression of Vegetation Evaporation Efficiency with Climate Wetness Index has ET/PET as the dependent variable and P/PET as the independent variable, with PET as the denominator of both variables. Such normalizing may influence the goodness of fit and perhaps the shape of the fitted function. I recommend redoing the analysis with just ET versus P as a cleaner test of the differences between genera.

   Response
   We concede the possibility that normalising on both axes may mask variation between genera. In Zhang et al (2004), the paper which inspired our analysis, several alternative models are proposed that relate VEE to CWI but all were normalised by PET. The rationale for this normalisation is that this places the models within the Budyko framework and so constrains evapotranspiration within both the energy and water limits imposed by climate.

   To provide more confidence in our conclusion regarding the two genera, we have tested for the effect of species on the parameters of two additional models: an exponential relationship between ET (evapotranspiration) and rainfall (P) and a linear relationship between ET/PET and P. These models and associated analysis of co-variance support our conclusion that the relationship between ET and climate is the same for the two genera. However, for the linear model the p-value for the species test is 0.155 or closer to significance than our original test. This results from the high leverage of two Eucalyptus sites on deep soils and another that received some supplementary irrigation during summer.

   All of this has provided a stronger basis for discussing the results and we have made the following amendments to the paper.
   i) Several extra lines of description in the Abstract (Lines 41 – 47)

ii) The last two paragraphs of the introduction have been changed to indicate that more than one model has been used to test for the effect of genus on evapotranspiration.

iii) Under section 2.5 (Meta-Analysis) the description of the analysis was amended to include the two additional models

iv) The most significant changes to the paper in response to this comment by Rev 1 can be found in the results section. These include an extra section 3.3 summarising the results (lines 408 and after) of the analysis and a Table 3 with significance values for the effect of species on each of the parameters. The full results of these additional analysis are included in the supplementary material as an extra sheet.

2. While differences in wood production between genera are briefly mentioned, some more discussion on this topic would be useful. If wood production per unit ET differed between genera, that would be an important consideration for forest management.  My expectation is that Eucalyptus would have a greater wood production per unit ET than Pinus.

   Response
   There are already several lines of discussion with specific reference to recent results in Chile and Australia (Lines 527-533).

3. It would also be useful to test if there are any differences between the different methods for estimating ET in the relationship between Vegetation Evaporation Efficiency with Climate Wetness Index or ET versus P.

   Response
   We repeated the analysis testing for 'methodology' and found no significant effect of methodology. These results can be found in the supplementary material.

4. The data spreadsheet included as Supplemental Material is useful but should have the column names defined and the data used in the paper identified specifically.

   Response
   We have provided a lot more information including column labels and comments within this spreadsheet. We have also provided an additional sheet within the worksheet that provided the results of the additional analyses.

Minor edits:

Response
All of the minor changes suggested have been made

L91: 'have found that not observed' should be 'have not observed'. (Corrected)

L144: Should 'Energy Limit' be included in the definitions? (Definition inserted in section 2.1.3)

L178: What is reference evaporation? (changed to potential evaporation to avoid confusion)

L341 and elsewhere: limit numbers to three significant digits.

L411: 'similar (Beynon and Doody, 2015), Figure 3)' should be 'similar (Benyon and Doody, 2015, Figure 3). (corrected)

Reviewer 2

I think this is a very useful contribution to the afforestation debate. The paper is very well written and presented. I do not have any particular criticisms. I note a posted comment regarding variability in species that, in an ideal world of data, would be good to delve into, but I suspect there are not the data to do this well. However it could be a discussion point.

Response
there is not sufficient data on individual species within genera for this analysis. As the reviewer notes his was also suggested in a community comment.

One section perhaps could be reworded; in the para beginning line 97 there is a statement that .."there seems to be differences between etc." and then suggests studies that show similar outcomes. Could be just me, but seemed not quite the right text.

Response
We agree and have made several changes to this paragraph that we hope improves clarity.

My other comments are around the Discussion, which I think is great. I suspect the section from Line 471-485 is the nub of the "thirsty eucalypts" concept that many have. What are people comparing when thinking about water use; is it mature closed canopy stand or a young stand going hard? What is the soil moisture status at the time of planting? etc.. with the pinus stands apparently having consistently

longer rotations, there is a greater likelihood of attaining a hydrologic state matched to the site resources.

There is a paper (Lane et al. 2005 H.Hydrol. 310) that compares changes in streamflow for mainly South Africa and Australian catchments. The lone eucalypt site plots right in the middle of the flow reductions. These magnitude of these reductions appeared to be partly a function of soil depth/storage. This analysis also looked at the timing of flow reductions which also speaks to the age and rototion discussion.

The authors may or may not feel like any of the above might be useful discussion material.

Overall, I commend the authors for a very good paper